# Chemoresistive Nanosensors Employed to Detect Blood Tumor Markers in Patients Affected by Colorectal Cancer in a One-Year Follow Up

**DOI:** 10.3390/cancers15061797

**Published:** 2023-03-16

**Authors:** Michele Astolfi, Giorgio Rispoli, Gabriele Anania, Giulia Zonta, Cesare Malagù

**Affiliations:** 1Department of Physics and Earth Sciences, University of Ferrara, 44122 Ferrara, Italy; 2SCENT S.r.l., Via Quadrifoglio 11, 44124 Ferrara, Italy; 3Department of Neuroscience and Rehabilitation, University of Ferrara, 44121 Ferrara, Italy; 4Department of Medical Sciences, University of Ferrara, 44121 Ferrara, Italy

**Keywords:** chemoresistivity, gas sensor, nanotechnology, VOCs, blood, colorectal cancer, follow-up, preventive screening

## Abstract

**Simple Summary:**

Since colorectal cancer represents one of the most diffused pathologies worldwide, usually lacking specific symptoms, it is crucial to develop and validate innovative low-invasive techniques to detect it. Here, a device based on an array of nanostructured gas sensors has been employed to analyze and discriminate the exhalations of blood samples collected from colorectal cancer-affected patients at different stages of their pre- and post-surgery therapeutic path. The device was clearly able to distinguish between the pre-surgery samples, where the tumor was present, and the one-year post-surgery ones, following the tumor removal. These results raise high hopes for the device’s clinical validation and its future use in clinical follow-up protocols, patient health status monitoring, and to detect possible post-treatment relapses.

**Abstract:**

Colorectal cancer (CRC) represents 10% of the annual tumor diagnosis and deaths occurring worldwide. Given the lack of specific symptoms, which could determine a late diagnosis, the research for specific CRC biomarkers and for innovative low-invasive methods to detect them is crucial. Therefore, on the basis of previously published results, some volatile organic compounds (VOCs), detectable through gas sensors, resulted in particularly promising CRC biomarkers, making these sensors suitable candidates to be employed in CRC screening devices. A new device was employed here to analyze the exhalations of blood samples collected from CRC-affected patients at different stages of their pre- and post-surgery therapeutic path, in order to assess the sensor’s capability for discriminating among these samples. The stages considered were: the same day of the surgical treatment (T1); before the hospital discharge (T2); after one month and after 10–12 months from surgery (T3 and T4, respectively). This device, equipped with four different sensors based on different metal–oxide mixtures, enabled a distinction between T1 and T4 with a sensitivity and specificity of 93% and 82%, respectively, making it suitable for clinical follow-up protocols, patient health status monitoring and to detect possible post-treatment relapses.

## 1. Introduction

Colorectal cancer (CRC) is the second most common diagnosed tumor in women (after breast cancer) and the third one in men (after lung and prostate cancers), representing about 10% of all the annual tumors diagnosed and deaths occurring worldwide. Thus, CRC is a very serious public health issue, in particular in the most developed countries [1] (e.g., Italy [2], where this study was performed). Given the lack of specific symptoms which could lead to a late clinical diagnosis, the planning of organized screening campaigns [3] is crucial for the identification of precancerous lesions before their malignant degeneration, and for their early treatment with proper therapies [4,5]. In Italy and in other countries, the most commonly adopted preventive screening technique for CRC is the fecal occult blood test (FOBT) in its two main versions: gFOBT (guaiac-based FOBT) and FIT (Fecal Immunochemical Test). In the case of a FOBT positive result, the patient undergoes colonoscopy afterwards, despite its invasiveness and dangerousness (it could cause a parietal lesion or even perforation). This technique is currently considered as the gold-standard for CRC diagnosis, because it allows a direct visualization of the colic mucous, the biopsy collection from suspect lesions and the eventual excision of small polyps [6,7]. On the other hand, the oncological five-year follow-up program for patients who have already ended CRC treatment is currently attained with a periodical clinical monitoring that however decreases the CRC-related mortality no more than 7–13% [8,9]. Based on the recommendation of the American Cancer Society and of the guidelines of the Associazione Italiana Oncologia Medica (CRC 2020 AIOM), the follow-up protocols, consisting in diverse progressive and invasive steps (e.g., antigen dosage sub-ministrations, abdomen computed tomography (CT), and colonoscopy [10]), are complex and very expensive, inducing a low subjects’ adherence to these examinations (about 50% [11]). On this basis, the research for specific CRC biomarkers and for an innovative method to detect them with low invasiveness is crucial to raise the patient adherence to oncological screening and monitoring programs [12]. It has been demonstrated that the cancer cells, because of their altered metabolism [13,14,15], discharge peculiar organic compounds into the blood stream that are volatile because of their high vapor pressure (called VOCs in the following) [16,17,18]. These VOCs originate by several processes occurring during the neoplastic degeneration such as cellular membrane peroxidation [13,14,15], an accelerated glycolysis activity [19], etc., and they could be therefore used as reliable tumoral markers. Currently, the VOCs’ detection and analysis have therefore a huge potential in the clinical practice, because they allow the development of effective screening and follow up in vitro techniques (with no or minimal invasiveness) for tumor prevention and monitoring [20,21,22,23,24,25,26,27,28,29,30,31,32,33,34,35,36,37,38,39]. To this aim, some studies employing gas chromatography–mass spectrometry (GC-MS), identified some VOCs with a good reliability which resulted in particularly promising CRC biomarkers [20,21,22,23,24]. Although the GC-MS technique is extremely powerful in detecting the single molecules composing the VOCs, allowing the identification of a cancer VOCs’ spectrum, it consists in a very expensive and bulky setup, hardly applicable in clinical practice, requiring a complicated data analysis and highly trained personnel [25]. In addition, the usage of the GC-MS has so far been limited to breath analysis [22,33,34,35,36,37] rather than on other human samples (such as blood, feces, etc.), according to the literature. Chemoresistive semiconductor gas sensors, realized at the Sensor Laboratory (SL) of the Department of Physics and Earth Sciences of the University of Ferrara (UNIFE), could be promising candidates to be employed in medical devices for CRC screening and monitoring. Based on many previous studies, these sensors have been demonstrated to be suitable for detecting VOCs at extremely low concentrations (up to ten parts per billion), in different human fluids and samples (such as stools, blood, intestinal tissues, etc.) [26,27,28,29,30]. The device employed here, called SCENT B2 in the following sections, is an upgraded version (in the electronics and in the management software) of two patented devices (SCENT A1 and SCENT B1 [40]), manufactured by the company SCENT S.r.l. [41] in a long-lasting collaboration with the SL. SCENT devices were able to detect the VOC pattern differences between the CRC-affected patients and the healthy controls, by examining blood, feces, biopsy and cell culture samples [26,27,28,29,30]. In particular, a previous feasibility study conducted using SCENT B1 provided very encouraging results in discriminating the blood samples collected from CRC-affected patients and the ones collected from presumed healthy subjects [27]. Given these promising results, the purpose of this study is to employ SCENT B2 to analyze the VOCs exhaled by blood samples collected from CRC-affected patients, at different stages of their pre- and post-surgery therapeutic paths [42].

## 2. Materials and Methods

### 2.1. The Sensors and the Device

A thick-film metal-oxide gas sensor was constructed of four independent parts: a sensing film, a substrate, a heater and a connector (here, a TO-39 socket). The first one consisted of a semiconductor film with a size of 1000 × 1000 × 30 µm, made of an interconnected network of nanograins (metal-oxide or sulfides) screen-printed (with serigraphic technology) onto an alumina substrate (the second part; 2.5 × 2.5 mm^2^), which had the dual function of electric insulator and mechanical support for the sensor. The sensing film can be synthesized by means of various approaches: here the sol–gel technique (which allows for the synthesis of nanostructures with the desired features, such as grain size, resistivity, etc., by choosing the proper process parameters) was employed to synthesize the sensing material [43,44]. The heater consists in a platinum coil printed on the substrate backside that heats the sensor to the optimal working temperature (called WT in the following; usually within 400–650 °C). Finally, the connector is a four pin TO-39 socket: two pins (Ø ≈ 0.06 mm) are welded to the heater and the others to the film through two gold comb-shaped electrodes previously printed on the substrate (an expanded view of the sensor is outlined in Figure 1). The welds were performed by thermo-compression, using a bonding machine.

The working principle of this sensor type consists in the variation in the sensing film resistivity as a consequence of the chemical redox reactions occurring between its surface and the gaseous molecules of the surrounding atmosphere (Figure 2) [45,46].

### 2.2. Chemical Section

#### 2.2.1. Sol–Gel Technique

The nanostructures at the basis of thick film MOX sensors consist in an interconnected network of nanograins with a size ranging between 50 and 200 nm. They improve sensibly the sensor performances (such as sensitivity, detection limit, recovery time, etc.) by maximizing the sensor active material surface to volume ratio, so promoting its interaction with the surrounding gaseous compounds. In order to obtain these nanostructures the sol–gel chemical process has been adopted for its many advantages in respect to other synthesis methods [43,47,48,49]. Besides the high costs of the raw starting materials, the main advantages of sol–gel are: low procedure costs, the possibility of obtaining homogeneous, stable and pure final products (also through low temperature treatments) and the full control of the final material’s features (such as nanostructures’ size, porosity, and so on) by carefully setting the process parameters. The sol–gel technique consists in the following four main steps:The hydrolysis of the precursor (as silicon or metal alkoxides) in water or alcohol solution (named “SOL”); to incentivize the SOL step; usually, some other chemicals (as acids, etc.) are added to the solution, in order to reach a colloidal suspension;The metal-oxide particles’ aggregation (or condensation), that occurs because of the water or alcohol removal from the solution and the metal–oxide bridging process arises. This phenomenon leads to the formation of a colloidal and viscous network of nanoparticles, but in a liquid phase, named “GEL”. At this stage, the solution alkoxide precursor and pH were the two main influencing parameters of the colloidal product; they can be carefully controlled in order to obtain nanoparticles with the desired size and cross-linking;The aging process, lasting up to a couple of days, during which several changes in the GEL structure and properties could occur (as the polycondensation). During this stage, the colloidal particle thickness increases while their porosity decreases. This is another sol–gel controllable parameter, in order to obtain a final product characterized by a certain porosity and grain size;The drying and calcination of the colloidal solution. Here, the GEL is dried at about 100 °C and then calcinated at higher temperatures (usually 400–800 °C). This step is crucial for the complete removal of the solvent residuals and/or of other chemical additives. During this process, the temperature and the relative humidity greatly influenced the quality of the final MOX nanopowder.

Finally, this nanopowder is converted in a printable viscous paste by adding a small amount of different organic vehicles (such as α-terpineol) and a glass frit (a glassy silicon oxides’ mixture). It is then deposited in form of a thick film (square 250 × 250 µm shape with a thickness of about 25–30 µm), on an alumina substrate between the two gold contacts by means of a screen-printing machine (mod. C920; Aurel s.p.a., Forlì-Cesena, Italy). The printed material is then again dried at about 100 °C and subjected to a firing treatment (usually from 650 to 850 °C).

Since the detectable gas spectrum of a single sensor is very wide, it is crucial to combine two or more different sensors into the same device (array) to significantly increase its selectivity (see Figure 2, Appendix A)). Based on the work carried out so far, the sensors selected for the present study, all heated to a WT of 450 °C, were:ST25 + 1%Au (or ST25): a mixture of tin and titanium oxide with the addition of 1% of gold nanoparticles (details about their synthesis and use are in [26,27,28,29,47]);SmFeO_3_: samarium and iron oxides [50];STN: tin, titanium and niobium oxides [51,52];TiTaV: titanium, tantalum and vanadium oxides [53].

#### 2.2.2. Sensor–Gas Interaction

The thick-film sensors based on the nanostructures synthesized with the sol–gel technique described above, are able to sense the gaseous compounds through the chemoresistive principle up to tens of ppb. This phenomenon requires firstly the sensor’s thermal activation (through thermionic effect), to maximize the number of electrons populating the conduction band, and thus the number of them able to overcome the grain–grain potential barrier (Band Bending, BB, in Figure 2). The other crucial role of the thermo-activation is the ionization of gaseous molecules in the atmosphere surrounding the sensor, promoting their adsorption on its surface. Hence, it is necessary to heat the sensor to an appropriate working temperature (WT), through the heater, usually selected on the basis of the best compromise between sensor response amplitude and its repeatability [52]. The MOX sensor works as follows: once it is placed in a reference atmosphere (i.e., pure dry air, pure CO, etc.), the surrounding oxygen ions and/or molecules (ionized at the WT) are adsorbed on the surface of each sensor grain. Then, the adsorbed oxygen ions, because of their electronegativity, act as acceptor surface states, trapping the conduction band (CB in Figure 2) electrons at the grain surface, affecting the sensor conductivity. As soon as the oxygen coverage ratio of the grain surface reaches the chemical equilibrium (i.e., when the number of adsorbed oxygens per unit of time is equal to the desorbed ones), the film conductivity becomes constant; this condition is called the baseline. For instance, in the case of an n-type sensor (based on n-doped semiconductor MOX material), the surface oxygen adsorption enlarges the grain “depleted shell” (i.e., the grain region lacking free charges), increasing the BB effect and consequently reducing the sensor conductivity. At this point, by subjecting the n-type sensor to a reducing gas (such as CO), the latter reacts with the previously adsorbed oxygen ions, releasing CO_2_ molecules in the atmosphere, freeing the trapped CB electrons: this decreases the grain–grain potential barrier (BB), increasing the sensor film’s conductivity (Figure 2). Once the reaction rate achieves a new equilibrium, the resulting constant film conductivity is the sensor signal in the gas presence and it is used to calculate the sensor response according to Equation (2). Finally, re-exposing the sensor to the initial reference atmosphere, the baseline condition is achieved in a so-called recovery time, which depends on a multitude of parameters, such as the sensor type, its WT, the tested gas kind and concentration, etc. The return of the sensor’s conductivity to its starting (reference) value demonstrates the full reversibility of the chemical reactions occurring on the grain surface [45,54,55].

### 2.3. The Sensor Response

The electronic circuitry connected to a sensor comprises the sensor heating and the sensor signal managing and acquisition. The first one adjusts the WT by controlling the voltage applied to the sensor heater and the current flowing through it [27,28,29]. The second one employs an operational amplifier in inverting configuration (Appendix A) returning a voltage $V_{O}$ (Equation (1)) as a function of time inversely proportional to the sensor film resistance:(1)$$V_{O}=-\frac{R_{f}}{R_{s}}V_{i}\;$$
where $R_{f}$ is the feedback resistor and $R_{s}$ is the sensor resistance. The sensor response R is calculated as the ratio between the voltage in the presence of a sample gas ($V_{Gas}$) and the one in the presence of a clean airflow ($V_{Air}$):(2)$$R=\frac{V_{Gas}}{V_{Air}}\;$$

*R* is therefore dimensionless and is independent from the measured physical quantity and the baseline amplitude (in general, different for each sensor) [29,56].

### 2.4. Blood Samples Collection and Analysis

This study is based on a wide observational, single center and prospective study, approved by the Ethics Committee on 13 July 2017 at the Surgery Department of Azienda Ospedaliero-Universitaria di Sant’Anna di Cona (Ferrara, Italy) with number 170,484. Thirty patients with a confirmed CRC diagnosis were recruited in the period between October 2020 and May 2022 at the S. Anna Hospital of Ferrara. The CRC-affected patients were randomly enrolled (i.e., regardless by sex, CRC stage, age, etc., as detailed in Table 1), after signing an informed consent for clinical data collection.

The patients’ inclusion criteria were:Age over 18 years old;CRC removal through laparoscopic or laparotomic elective surgery.

The patients’ exclusion criteria were:
Pregnant women;Emergency surgical treatment.

Blood samples were gathered at different moments of the patients’ therapeutic path at the Hospital S. Anna as follows:T1: the same day, but before the surgical treatment;T2: before the hospital discharge (with non-standardized timings, depending upon the patient clinical course);T3: after at least one month after surgery (organizing a return of the patients to the hospital);T4: after 10–12 months from surgery (organizing a second return of the patients to the hospital).

### 2.5. Experimental Section

Blood samples were collected at the Ferrara Hospital (S. Anna, Cona) and stored at room temperature in 7 cc vials containing the anticoagulant agent K3-EDTA, in order to inhibit blood clotting and to guarantee their correct conservation. The samples were then transferred within one hour of storage to the Electrophysiology Laboratories of the Department of Neurosciences and Rehabilitation of UNIFE for the VOC pattern analysis with the sensing device SCENT B2. The clean and dry air flux (see Device in Appendix A) was initially conveyed directly to the sensors to set a stable baseline. Once it was ensured that the baseline was stable, the blood vial was poured in a reusable Teflon container (previously washed with ultrapure water and dried with a sterilized air flux) and placed immediately in the SCENT B2 sample box by the operator (Appendix A). This procedure was executed as rapidly as possible in order to minimize external air contamination and possible VOCs’ loss. At this point, by turning the three-way-valve, the clean and dry air flux was directed through the sample box (thus gathering the headspace chemicals exhaled by the blood sample) before reaching the sensors; the latter, interacting with VOCs, generated the responses. Once the sensor signals attained a plateau, the three-way-valve was turned to the initial position to convey the air flux toward the sensors, in order to return their signals to the initial baseline. Finally, the blood sample was properly disposed of and the Teflon container meticulously washed with bleach and several times with ultrapure water, and finally dried with a sterilized air flux. The data stored as .txt files were also plotted off line as the response R in function of time (Equation (2) and Appendix A) and further processed through different statistical techniques, such as principal components analysis (PCA) and Receiver Operating Characteristic (ROC). The accuracy of discrimination between two sample types of each sensor and of the entire device was evaluated through their sensitivity (i.e., the ability to recognize true positives), and their specificity (i.e., the ability to recognize true negatives).

## 3. Results and Discussion

### 3.1. Ensemble Statistical Analysis and After-Surgery Follow-Up

The blood of thirty CRC patients was collected at four different stages of their surgical clinical path: the same day or one day before surgery (T1); just before the patient discharge (T2); after one month from surgery (T3); and after 10–12 months from surgery (T4), for a total of 109 blood samples (instead of 120, because some patients were not able to provide all the four samples for several reasons such as early dehospitalization, death, etc.). Surprisingly, T2 gave larger responses than T1 (Figure 3 and Table 2), probably because of the patients’ increased catabolism following the surgery, and of the VOCs that can be potentially produced by drugs and anesthetics used during the surgery or VOCs produced by their metabolites. Indeed, it has been found that the sensor responses to the blood collected from all the patients (but unfortunately only four of them) just before the surgery were always significantly smaller than the one collected after the anesthesia, but before the tumor removal. Moreover, another limitation in the T2 analysis is given by the unavoidable non-standard collection time of the blood samples, due to the different patient discharge times. Finally, many patients were subjected to adjuvant therapies, which may contaminate the blood exhalation and detected by the sensors.

For all the above reasons, and because this study was focused on the patient follow-up after surgery, to evaluate the sensor performance in distinguishing the exhalations of a blood sample collected from a patient affected by tumor (T1) and the ones after one year from its surgical removal (T4), the statistical analysis was focused on T1 and T4 only. The sensor responses R to the blood samples at T1 and T4 for 28 out of 30 patients enrolled in this study are reported in Figure 4 (unfortunately, two patients provided the T1 sample only because they could not return for the annual control, when the T4 was collected). None of the patients was subjected to neoadjuvant therapies (i.e., chemotherapy, radiotherapy, etc.) and adjuvant radiotherapy, but adjuvant chemotherapy was performed on eleven out of twenty-eight patients. Since these therapies could affect the sensor’s response reliability and repeatability (due to possible chemotherapy compounds and/or their metabolites discharged in the blood stream), the homogeneity was assessed (i.e., if two data groups belong to the same statistical distribution, according to the null hypothesis for which the averages of the two populations are the same) of the two T4 groups of sensor responses: the samples collected from patients subjected to adjuvant chemotherapy ($n_{u}=17$) and the untreated ones ($n_{t}=11$). This homogeneity was assessed by performing a unpaired two-sample Student’s *t*-test on the two independent sample groups: the Student’s *t*-test was chosen because of the number of data, n=28, and the unknown population variances. The test variable t for each sensor resulted to be: 0.189, 0.467, 0.017 and −0.420 for the ST25, SmFe, STN and TiTaV sensor, respectively. Since the critical value $t_{c}\left(a,g\right)=3.067$ (significance level: α = 0.01; degrees of freedom: $g=n_{u}+n_{t}-2$), retrieved from the Student’s *t*-test bilateral distribution table, was much larger than the above *t*-test variables (related to the sensor responses), then they fell into the area of the Student’s *t* distribution in between $-t_{c}$ and $+t_{c}$. Therefore, the null hypothesis was confirmed and consequently the T4 dataset was independent from the adjuvant chemotherapy treatment.

The sensor responses R to the T4 samples were generally smaller than the T1 ones (Figure 4, four upper panels), as is more clearly shown by the R percentage reduction in T4 in respect to T1 for each sensor and patient (Figure 4, bottom panel). This result is expected, since the surgical removal of the tumoral mass would eliminate the cancer cell VOCs dumped in the blood stream.

A clear statistical difference between T1 and T4 readily emerged in the average response of all sensors: in particular, it decreased by 21% for STN (that demonstrates to be the most suitable sensor here), 16% for ST25 − 1%Au, 8% for TiTaV and 7% for SmFeO_3_ (Figure 5). The suitability of the STN sensor was expected on the basis of previous results obtained by our research team in feasibility studies on blood samples [27,30]. To assess the statistical significance differences between T1 and T4 for each sensor, a paired two-sample *t*-test (α = 0.01; g = 26) was performed. By comparing the t variables (that resulted: 4.458, 4.317, 6.004, and 5.588 for ST25, SmFeO_3_, STN and TiTaV, respectively) with the $t_{c}$ retrieved from the Student’s *t* bilateral table, it results in clear statistical differences between the T1 and T4 for all sensors (*p*-values < 0.001 for ST25 and SmFeO_3_, and *p*-values < 0.0001 for STN and TiTaV; Figure 5).

This difference is further highlighted by the significant statistical separation between the T1 and T4 populations by the confidence interval analysis performed by using the Student’s *t* distribution (*n* = 30 for T1 and *n* = 28 for T4) with a 95% confidence interval (significance level: α=0.05; Figure 6).

### 3.2. Sensor Accuracy Evaluation

In order to evaluate the accuracy of each sensor in discriminating T1 and T4, the sensitivity and the specificity parameters were computed as follows:(3)$$Sens.=\frac{TP}{TP+FN}\;;\;Spec.=\frac{TN}{TN+FP}$$
where TP, TN, FP and FN are the number of true positives, true negatives, false positives and false negatives, respectively. An extensively used methodology to evaluate the accuracy of a medical screening test is provided by the ROC (Receiver Operating Characteristic) analysis [57,58], consisting in the plot of Sens. vs. 1-Spec. (Figure 7). The sensitivity, the specificity, the AUC and the cut-off amplitude for each sensor are reported in Table 3.

### 3.3. Sensor Discrimination Power

To better assess the discriminating power between T1 and T4 of the four sensors, their responses were further analyzed with the Principal Component Analysis (PCA) [59]. The four eigenvectors of the covariance matrix (PC1, PC2, PC3, PC4), calculated starting from the four-dimensional plot of the sensor responses, were plotted one vs. the other to construct the two-dimensional score (or dispersion) plots.

The fraction of variance (in percent) of the datasets projected on the PCs were: 96% for PC1, 2.4% for PC2, 1.4% for PC3 and 0.2% for PC4. Since the PC1 vs. PC2 plot contains most of the dataset information, having the largest fraction of the total variance (about 98.4%), PC3 and PC4 (having a total variance of 1.6%) were neglected. The distribution of the red points (T1) and the blue ones (T4) are enough discriminated, giving a small discrepancy between the T1 and the T4 data (Figure 8). The ROC curve calculated on the PC1 projected dataset (i.e., the dataset space direction where its variance is maximized) is characterized by an AUC and a cut-off amplitude of 0.93 and 0.84, respectively (Figure 9 and Table 3). The latter values arise from the contribution of all the four sensors and not from a single one (as reported in Figure 7); therefore, they represent the performance of the device as a whole, making it suitable for a future use in the clinical practice.

## 4. Conclusions

The SCENT B2 equipped with STN, TiTaV, ST25 − 1%Au and SmFeO_3_ sensors, reliably discriminates the blood VOC pattern changes between T1 (collected just before surgery) and T4 (collected after 10–12 months from surgery) with a sensitivity of 93% and a specificity of 82%. This excellent performance makes SCENT B2 a very suitable candidate for its future development as a medical device to be used in clinical follow-up, to test the patient’s health status, and to detect possible future relapses after the surgical treatment. The future goals are:The recruitment of a larger number of patients in order to validate SCENT B2 as a medical device;The further development of SCENT B2 to embed the host computer in the device by using a Raspberry Pi board;The improvement of the sensor technology in order to better detect the CRC stages and to extend the use of this device to other tumor types;The undertaking of a clinical trial in order to validate the device as a diagnostic equipment.

## 5. Patents

C. Malagù, S. Gherardi, G. Zonta, N. Landini, A. Giberti, B. Fabbri, A. Gaiardo, G. Anania, G. Rispoli, L. Scagliarini, Combinazione di materiali semiconduttori nanoparticolati per uso nel distinguere cellule normali da cellule tumorali (2015), National #: 102015000057717.

## Figures and Tables

**Figure 1 cancers-15-01797-f001:**
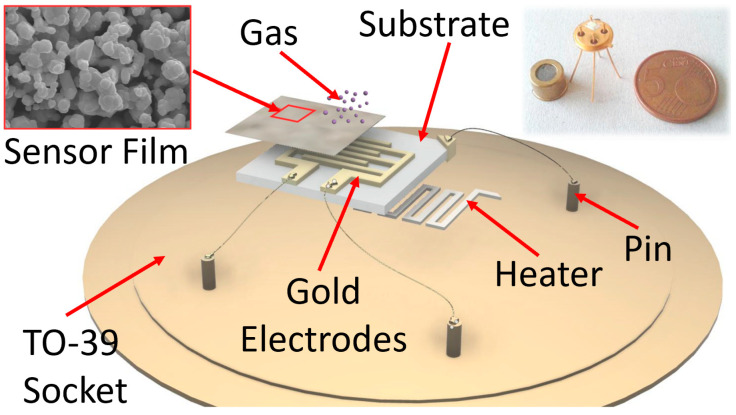
Sketch of a sensor. The film reacting with the gas is screen-printed onto the alumina substrate, endowed with gold comb-shaped electrodes; the heater is placed on the alumina substrate backside. Inset on the left: SEM image of the nanostructured shape of the sensing film material (scale bar is 2 μm); inset on the right: photo of a real sensor compared with a EUR 5 cent coin.

**Figure 2 cancers-15-01797-f002:**
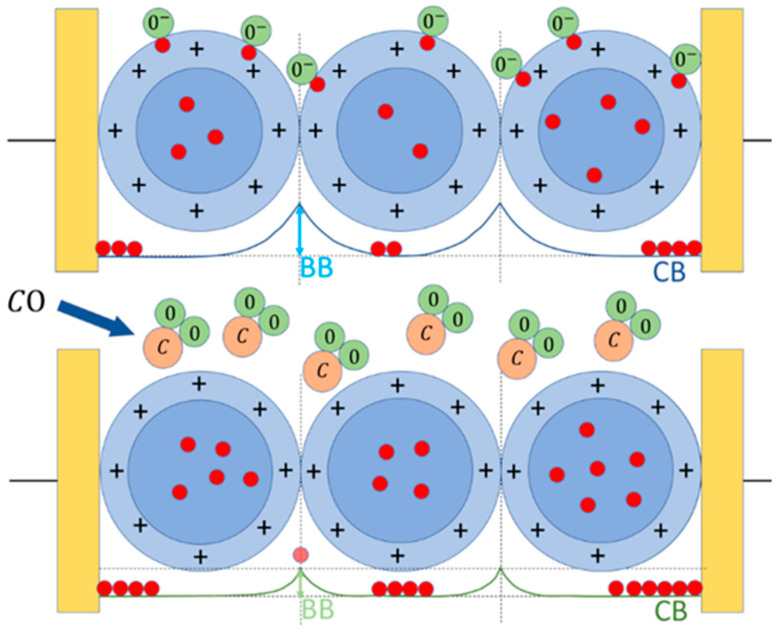
Working principle of an n-type sensor. Upper panel, the sensor grains (blue circles) placed in a reference atmosphere (here, dry air) interacting with the oxygen ions and/or molecules (green circles), which entrap the electrons (red circles) at the sensor grain surface (the external border of the light blue annulus), subtracting them from the grain bulk (blue circle), enlarging the depletion region (light blue annulus) and consequently increasing the grain–grain potential barrier (BB). Lower panel, sensor exposed to CO (C is light brown circles): the oxygen atoms adsorbed at the grain surface reacts with CO, releasing CO_2_ in the environment and giving back electrons to the bulk CB, reducing the BB effect.

**Figure 3 cancers-15-01797-f003:**
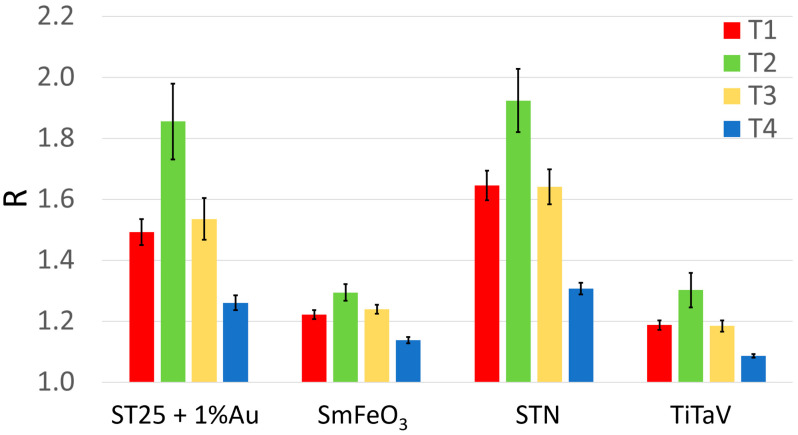
Average sensor responses (R) to the blood collected during patient follow-up. Response R averaged on all patients and relative standard error (reported numerically in Table 1) of each sensor at the different blood collection times (red, T1, *n* = 30; green, T2, *n* = 22; yellow, T3, *n* = 29; blue, T4, *n* = 28).

**Figure 4 cancers-15-01797-f004:**
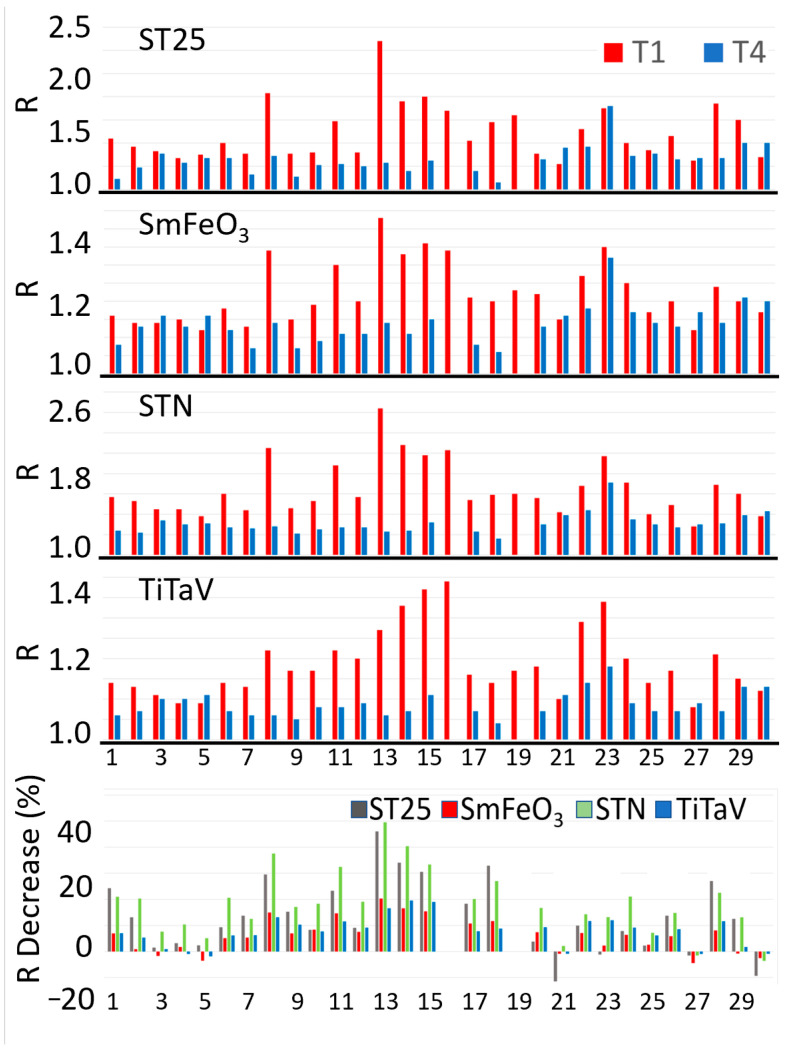
After-surgery follow-up of 28 patients. The four upper panels show the R amplitudes for each sensor at T1 and T4 times for each patient. The bottom panel shows the R percentage reductions in T4 in respect to T1 for each sensor and patient. The numbers in abscissa in all panels represent the randomized patient numbers; the patients n. 16 and n. 19 of the bottom panel are absent because they provided the T1 sample only (see text).

**Figure 5 cancers-15-01797-f005:**
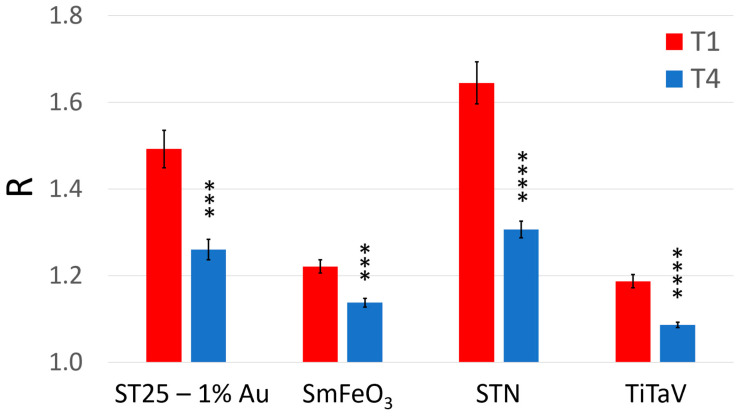
Average sensor responses (R) to the blood collected just before surgery and after about one year of follow-up. Response R averaged on all patients and relative standard error (reported numerically in Table 1) of each sensor at the two blood collection times (red, T1; blue, T4). The statistically significant differences between T1 and T4, for each sensor, have been calculated with a significance level α = 0.01 (*** *p* < 0.001; **** *p* < 0.0001).

**Figure 6 cancers-15-01797-f006:**
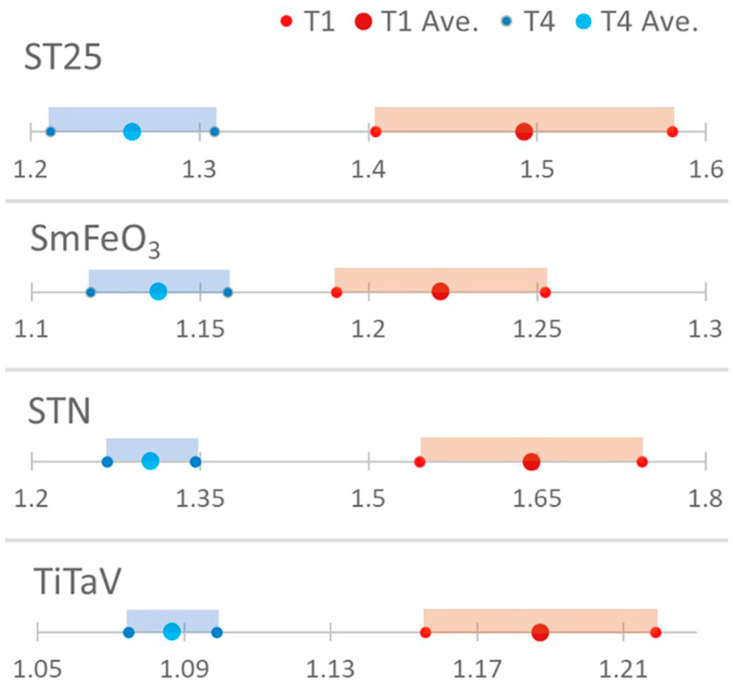
Confidence interval analysis. The statistical discrimination between T1 and T4 was evaluated by computing the confidence intervals by using the Student’s *t*-test (*n* = 30 for T1 and *n* = 28 for T4).

**Figure 7 cancers-15-01797-f007:**
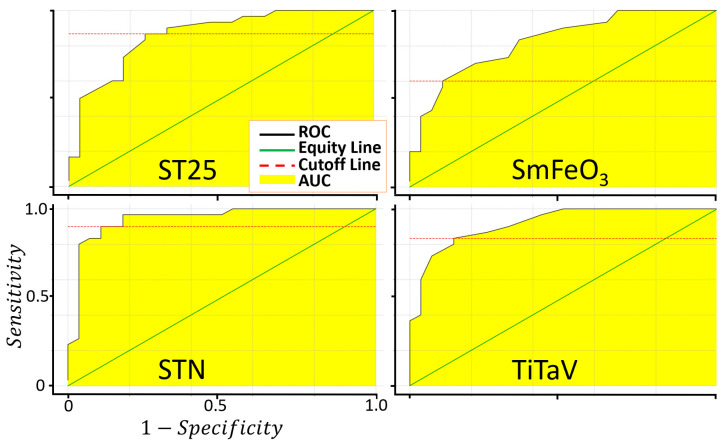
Evaluation of the sensor sensitivity and specificity in discriminating between T1 and T4. Plots of the ROC curves of the four sensors resulting from 1000 iterations.

**Figure 8 cancers-15-01797-f008:**
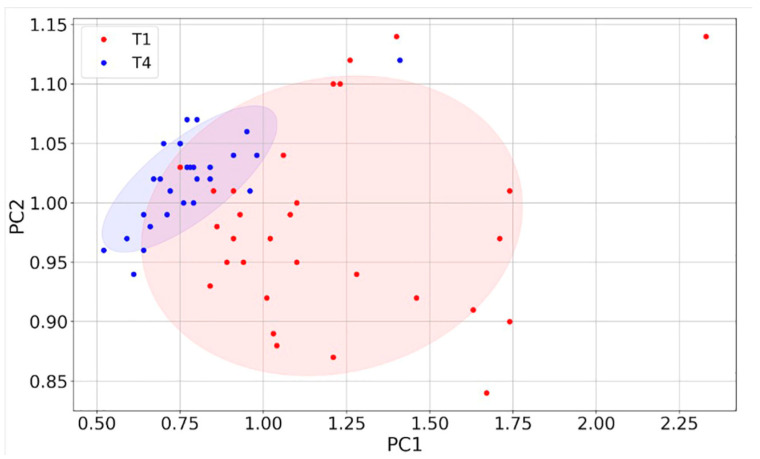
PCA of sensor responses. Score plot of PC1 vs. PC2 (red points: T1; blue points: T4) constructed with the responses (*n* = 58) of the four sensors.

**Figure 9 cancers-15-01797-f009:**
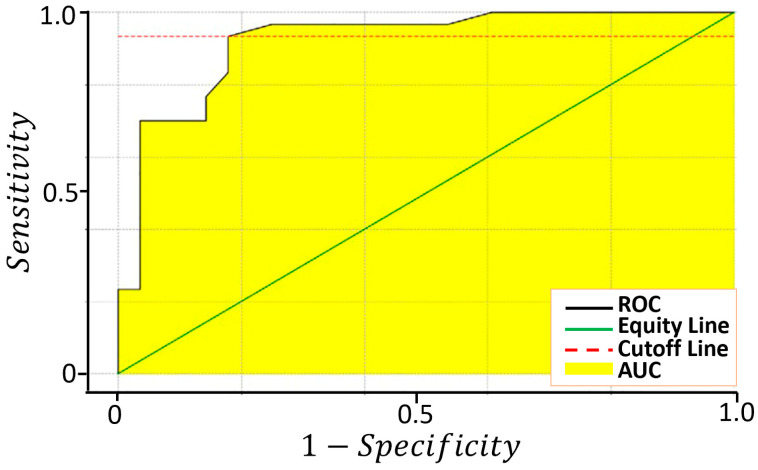
Evaluation of the sensitivity and specificity of the device in discriminating between T1 and T4 samples. ROC curve calculated on the PC1 projected dataset resulting from 1000 iterations.

**Table 1 cancers-15-01797-t001:** The composition of the randomized enrolled population. AVE. AGE is the average population age; BMI is the body mass index.

Enrolled Population			
Sex	Male	19	63%
	Female	11	37%
Ave. Age	Male/Female	69	47–87
BMI	>30	9	30%
	<30	21	70%
Tumor Localization	Ascending Colon	18	60%
	Transverse Colon	3	10%
	Descending Colon	3	10%
	Sigmoid	3	10%
	Rectum	3	10%
Stage	I	3	10%
	II	13	43%
	III	13	43%
	IV	1	3%

**Table 2 cancers-15-01797-t002:** Average responses R ± their standard error of the four sensors to T1, T2, T3 and T4 of Figure 3.

	ST25+1%Au	${\mathrm{SmFeO}}_{3}$	STN	TiTaV
${\bar{R}}_{T1}$	1.49 ± 0.04	1.22 ± 0.02	1.64 ± 0.05	1.19 ± 0.02
${\bar{R}}_{T2}$	1.85 ± 0.12	1.29 ± 0.03	1.92 ± 0.10	1.30 ± 0.06
${\bar{R}}_{T3}$	1.54 ± 0.07	1.24 ± 0.01	1.64 ± 0.06	1.18 ± 0.02
${\bar{R}}_{T4}$	1.26 ± 0.02	1.14 ± 0.01	1.31 ± 0.02	1.09 ± 0.01

**Table 3 cancers-15-01797-t003:** Main parameters characterizing the sensor and the device performances. The first four columns shows the AUC, cut-off, sensibility and specificity for each of the four sensors calculated from the ROC curves of Figure 7; the last column shows the parameters derived from the ROC curve (calculated on the basis of the PC1 data projection of Figure 8) globally characterizing the new device equipped with the four sensors (Figure 9).

	ST25	SmFeO_3_	STN	TiTaV	PC1
AUC	0.87	0.84	0.94	0.93	0.93
Cut-Off	1.29	1.18	1.39	1.11	0.84
Sensitivity	87%	60%	90%	83%	93%
Specificity	75%	89%	89%	86%	82%

## Data Availability

Data are available under request.

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
