# Peer review of "Chemoresistive Nanosensors Employed to Detect Blood Tumor Markers in Patients Affected by Colorectal Cancer in a One-Year Follow Up"

_cancers, 2023, doi:10.3390/cancers15061797_

Round 1

Reviewer 1 Report

Comments: Few critical points should clarify before being accepted.

1-    The manuscript lacks information about how the interaction between the sensor and serum could be done. the chemical recognition was not described if the sensor is sensitive to specific protein, antigen or molecule in serum of patients

2-    Chemical section was missed in section " Materials and Methods"

3-    A scheme or photograph should to be added to illustrate  the structure of sensors

4-    It was written in line 205 "T2 gave larger responses in respect to T1" This was explained by authors that could be due to the patient inflammatory state. This explanation gives me some doubts if sensors are sensitive to inflammatory cytokines and If  the blood examination was done for patients in T4  to confirm there was no inflammation

5-    In (figure 1, 3) R here is the percentage or what is it distinguished?

Reviewer 2 Report

The manuscript describes a device to detect volatile organic compounds originated in colorectal tumor. The overall aim of the study is interesting; however, I have following comments that should be addressed;

1. Composition of VOC is missing. VOCs identification is important to mark them as biomarkers. I think only using VOC is not an advancement.

2. Results may be validated using GC-MS after identifying VOC.

3. Healthy person's blood should be used as a control for comparison. I cannot find any data about that.

4. Authors should reconfirm conflict of interest, if any due to patent as list of inventors is different from authors.

5.  Are VOCs only related to colorectal cancer?

Round 2

Reviewer 1 Report

The manuscript was revised point by point according to reviewer comments 

Reviewer 2 Report

Authors have reasonably answered my comments.